# Axial Stress Measurement of Steel Tubes Using Ultrasonic Guided Waves

**DOI:** 10.3390/s22093111

**Published:** 2022-04-19

**Authors:** Siyuan Zhu, Xian Xu, Jinsong Han, Yaozhi Luo

**Affiliations:** 1College of Civil Engineering and Architecture, Zhejiang University, Hangzhou 310058, China; zvxzvx4@gmail.com (S.Z.); luoyz@zju.edu.cn (Y.L.); 2Center for Balance Architecture, Zhejiang University, Hangzhou 310058, China; 3College of Computer Science and Technology, Zhejiang University, Hangzhou 310027, China; hanjinsong@zju.edu.cn

**Keywords:** steel tube, guided wave, stress measurement, acoustoelastic, torsional mode

## Abstract

Axially loaded steel tubes are widely used as primary structural members in civil engineering structures. In this paper, a stress measurement method for axially loaded steel tubes is developed based on the linear relationship between the group velocity of guided waves in the steel tube and the stress of the steel tube. The propagation modes of guided waves in a typical steel tube are analyzed using semi-analytical finite element method. A torsional mode T(0,1) is adopted to conduct the measurement. Experiments are carried out to calibrate the linear relationship between the group velocity of guided waves in a steel tube and the stress of the steel tube. The calibrated linear relationship is verified by another round of experiments on the same steel tube specimen. There is an average error of 8.2% between the stresses predicted by the calibrated linear equation and those obtained from strain gauges. Via this study, the guided wave-based stress measurement method has been successfully extended to axially loaded steel tubes.

## 1. Introduction

Steel tubes are widely used as structural members in the area of civil engineering, especially for axially loaded members. The in-service stress level of a steel tube member may deviate from its design level due to unexpected loads, stress redistribution, structural damages, etc. The absolute stress in a steel tube member is an essential parameter for safety evaluation of the member and the structural system. However, few methods are able to detect the stress level of an in-service steel tube member which is under load before the detection.

There are various approaches for non-destructive stress measurement of steel members, such as resistance strain-gauge transducer [1], optical fiber sensor [2] and vibrating wire sensor [3], X-ray diffraction method [4], magnetoelastic method [5], and ultrasonic method [6]. However, some of them, including the strain-gauge, optical fiber sensor, and vibrating string sensor, must be pre-deployed on the stress-free member if they are planned to measure the absolute stress of the member in service. In other words, these approaches are not able to gain the absolute stress of an in-service member without pre-deployment of the sensors. For the other approaches, including the X-ray method, magnetoelastic method, and ultrasonic method, pre-deployment is not required. The X-ray diffraction method relies on the interaction between the X-ray beam and the crystal lattice of the material [4]. This method possesses high precision in laboratory measurements, yet it is susceptible to the quality of the component surface. Besides, the equipment is large, complex, and expensive, making it inconvenient for field measurements [7]. The magnetoelastic method is capable of making non-contact measurements, and its equipment is less complex. However, due to the need for the magnetization process, its application is limited by the magnetization conditions [8]. The ultrasonic method is based on the acoustoelastic theory [9,10], which provides a relationship between the stress and the velocity of elastic waves in solids. By using this relationship, the ultrasonic can be applied to measure the residual stress in welded regions [11], and the axial stress of high strength bolts [12], rails [13], and multi-wire strands [14]. This method is fast and simple in operation, and its equipment is low-cost and portable. Its disadvantage is that the acoustoelastic effect is a weak effect, and thus customized methods are required to ensure its feasibility in specific applications.

Guided waves are a kind of ultrasonic waves which propagate over the entire cross-section of the components. Its low attenuation and high propagation distance make it possible for long-range and hidden-area detection. Gazis [15] put forward the dispersion equation of guided waves in homogeneous and infinite tubes. Silk and Bainton [16] experimentally investigated the generation of guided waves in thin-walled metal tubes and labeled the different wave modes. Alleyne and Cawley [17,18] developed a dry-coupled piezoelectric transducer system to excite the L(0,2) mode and suppress all the non-axisymmetric modes for long-range detection. Due to the potential in in-situ testing, guided waves have been generally adopted in defect detection [19] and stress measurement [20] in recent years. Chen et al. [21] used the first order longitudinal guided wave to detect the axial stress of a prestressed steel bar and found a linear relationship between the group velocity of the first order longitudinal guided wave and the stress of the bar in the low frequency domain. Scalea et al. [22] excited longitudinal guided waves in seven-wire strands and pointed out that the velocity–stress relationship is similar to the acoustoelastic trend at high stress level but just the opposite at low stress level. Chen and Wissawapaisal [23] revealed that the time of flight (TOF) of guided waves have a linear relationship with the stress in steel strand when the axial force ranges between 18% and 70% of the ultimate strength of the steel strand. Washer and Green [24] evaluated the acoustoelastic effects in prestressing tendons and then designed non-contact electromagnetic acoustic transducers for the launch and reception of guided waves. Gandhi et al. [25] developed the theory of acoustoelastic Lamb wave propagation for isotropic plates subjected to a biaxial, homogeneous stress field. Liu et al. [26] discussed the evolution of the missing frequency band of guided waves in slightly tensioned steel strands and developed a new tensile force measurement method capable of measuring incremental stress of approximately 3 MPa. Loveday [27] investigated the measuring method for one-dimensional temperature stresses and axial forces in continuously welded rail. Dubuc et al. [28] explored the effect of axial stress on higher order longitudinal guided modes propagating in individual wires of seven-wire strands. Wu et al. [29] presented a finite element method using eigenfrequency to analyze the wave propagation in prestressed waveguides. Yang et al. [30] took temperature into account and proposed a thermo-acoustoelastic theory combined with the semi-analytical finite element method to investigate thermal effects on acoustoelastic guided wave propagation. In the above studies, a linear relationship between the velocity of guided waves and the stress level was examined under some conditions. Still, nonlinearity between them was also observed in some situations [26,31,32]. In general, most previous studies focused on steel rods and strands. Seldom were studies on guided wave-based method for stress measurement of steel tubes reported, to the authors’ knowledge.

As mentioned above, for civil structures, the axial stress measurement of steel tubes is of great demand, and the guided wave-based method has the potential to meet the demand. In this study, a guided wave-based method for axial stress measurement of steel tubes is proposed and experimentally validated. The implementation of the method is customized according to the features of steel tubes. In Section 2, the theoretical basis of the proposed method is clarified. Based on the guided wave theory, the requirements of the stress measurements in tubes are discussed, and the appropriate wave mode for excitation is selected. In Section 3, the measurement methodology fit for tubes, including equipment arrangement and signal processing method, is presented. In Section 4, experiments are carried out to calibrate the linear relationship between the stress and the group velocity of guided waves in steel tubes. The feasibility of the calibrated linear relationship is verified by another round of experiments on the same specimen.

## 2. Theory

### 2.1. Acoustoelastic Theory under Axial Stress

According to the acoustoelastic theory [10], in a homogeneous and isotropic infinite solid subjected to uniaxial stress, the velocity of ultrasonic waves propagating in the same direction as the applied stress can be written in the first-order approximation [33] as
(1)$$V_{L}=V_{L}^{0}(1+K_{L}\sigma )$$
(2)$$V_{T}=V_{T}^{0}(1+K_{T}\sigma )$$
where *V_L_* and *V_T_* are the velocities of longitudinal and transverse waves, respectively; *σ* is the applied stress; *V_L_*^0^ and *V_T_*^0^ are the longitudinal wave velocity and the transverse wave velocity when the stress *σ* = 0, and they are determined by the material properties of the solid as given in Equation (3); and *K_L_* and *K_T_* are the acoustoelastic coefficients representing the response of the material to the stress and propagating waves, and they are also determined by the material properties as given in Equation (4) [33]. This approximation holds in most situations, since the change in wave velocity is much smaller compared to that in stress.
(3)$$V_{L}^{0}=\sqrt{\frac{\lambda +2\mu }{\rho }},\;V_{T}^{0}=\sqrt{\frac{\mu }{\rho }}$$
(4)$$K_{L}=\frac{\left(\lambda +\mu \right)\left(4\lambda +10\mu +4m\right)∕\mu +\lambda +2l}{2(\lambda +2\mu )(3\lambda +2\mu )},\;K_{T}=\frac{4\lambda +4\mu +m+\lambda n/4\mu }{2\mu (3\lambda +2\mu )}$$
where *ρ* is the density of the material, *μ, λ* are the Lamé elastic constants, and *l, m, n* are the Murnaghan’s third-order elastic constants.

Equations (1) and (2) illustrate that for a given material, the *V_L_* and *V_T_* are linear to the axial stress. Besides, it is numerically and experimentally proved that this relationship still holds for most waveguides [21,22,27,29,33,34], Although the material parameters in Equations (3) and (4) are not easy to be obtained precisely, the coefficient *K* can be experimentally determined by measuring the wave velocity under several given stresses. Next, the equations can be used to determine the stress in the same material by measuring the wave velocity in it. 

### 2.2. Guided Waves in Tube

When ultrasonic bulk waves propagate under the limitation of a solid’s boundary, the guided waves generate. They will continuously interact with the boundary, such as reflecting and refracting. The corresponding solid is called a waveguide whose size in the dimension orthogonal to the direction of propagation is always smaller than the wavelength of the guided waves. Compared to the bulk waves, guided waves propagate over the whole section of the waveguides, and the solution of its wave equation needs to satisfy additional boundary conditions. In a tube (i.e., a hollow cylinder) there are three groups of modes: longitudinal mode, torsional mode, and flexural mode, labeled as L(0,*m*), T(0,*m*), and F(*n*,*m*), where *n* is the harmonic order and *m* is a sequence number. They are distinguished from each other by the vibration patterns of the particles. The longitudinal mode and the torsional mode are axially symmetrical, while the flexural mode is not.

The dispersion curves describe the behavior of the guided wave, and for tubes it can be obtained by solving the dispersion equation given by Gazis [15]. The solution of the equation consists of a real number and imaginary number. The real number part of the solution is used to plot the dispersion curves, and the imaginary part is related to dissipation which is not concerned here for its insignificant influence on stress measurement. In recent years, other methods such as finite element method, semi-analytical finite element method, and boundary element method are also applied to obtain dispersion curves by numerically solving the wave equation [35]. In this study, semi-analytical finite element (SAFE) method [36] is used to obtain the dispersion curves of a steel tube with given sizes. In the SAFE method, the wave motions in the propagation direction z are theoretically described by harmonic function exp(i*kz*-i*ωt*), where *k* is wavenumber and *ω* is circular frequency, and the x-y cross-section is discretized. The governing equations of the discretized system are given by the virtual work principle. By numerically solving the equations, the dispersion curves can be obtained. For example, the dispersion curves of a steel tube with a diameter of 88.5 mm and a thickness of 4.0 mm, which would be used in the experiment, can be obtained by SAFE method, as shown in Figure 1. Note that material density *ρ* = 7850 kg/m^3^, Young’s modulus = 206 GPa, and Poisson’s ratio = 0.3 are assumed for the steel. 

For each mode, the group velocity varies with frequency, which is called frequency dispersion. It will naturally cause distortion of the wave packet in propagation. In the real world, the frequency may also be influenced by the environment or other factors, so the wave velocity changes as well. 

As mentioned above, to detect the axial stress in the tube, the wave velocity is required to be determined. In practice, the propagation distance and the corresponding TOF are usually easier to be directly measured, and then the wave velocity can be determined by the ratio of them. As a result, the wave velocity determined in this way is actually an average velocity. If the current velocity changes because of frequency dispersion, it will lead to a variation in measured TOF and cause an error that is difficult to recognize. Therefore, it is better to choose a mode in which the dispersion effect is insignificant. The L(0,2) and T(0,1) modes are first considered for the given tube. Because both of them have a considerable large section where the velocity changes little with the frequency, they are also easy to be excited and received. Note that frequency higher than 200 Hz is not considered in this study in order to excite fewer wave modes, making it easier for wave signal identification.

Most previous studies prefer longitudinal modes because it is more sensitive to stress [13], i.e., the change of the wave velocity to a given stress increment will be more significant. In another aspect, it is difficult to excite torsional modes in slender members such as rods and strands in the experiments, and thus the torsional modes are usually ignored. However, for tubes considered in this study, the torsional mode is seriously considered and is found to have some additional advantages for the experiment. Firstly, the first-order torsional mode theoretically has no frequency dispersion [37], which means the wave packet changes less in propagation. It is beneficial for the accuracy of the signal identification and the determination of TOFs. Secondly, the velocity of the T(0,1) mode is equal to that of the body transverse wave, which is only related to the material. Thus, the sectional size of the tube will not affect the experiment results. In future applications in members with the same material but different sections, repeated acoustoelastic coefficient calibrations may not be required. The last advantage of it is that the velocity of T(0,1) mode is much lower than that of L(0,2) mode and thus will have a larger TOF for a given propagation distance. It will lead to a smaller relative error in the experimental measurement. As a result, the T(0,1) mode is finally chosen in this study.

## 3. Experiments

### 3.1. Equipment Setup

To carry out an experiment to validate the proposed method, a scheme, as shown in Figure 2, is devised. Axial forces are applied on both sides of a straight tube to produce uniform axial stress. The parameters of the guided wave of T(0,1) mode are set in a program and sent to the ultrasonic testing machine. The transmitting transducer located at one end of the tube converts the electrical signals into circumferential displacement to excite the expected wave. It propagates axially and is received by two transducers located at a given distance *L* from each other. By identifying the times when the two transducers receive the same wave packet, the TOF for the distance *L* can be obtained. 

The equipment arrangements corresponding to the scheme are shown in Figure 3. A jack (ESH-206, Eagle Pro) with a hydraulic pump is installed on a steel frame to apply forces on the steel tube specimen. A compression force sensor is used to measure the applied axial force. Two transverse wave piezoelectric probes with a diameter of 20 mm are selected as the receiving transducers, and a magnetostrictive transducer composed of a metal strip and a coil is selected as the transmitting transducer. They are tightly attached to the specimen by epoxy adhesive. At the location of each receiving transducer, four strain gauges are evenly distributed around the circumference to crosscheck the stress in the specimen. The strain gauges and transducers are well staggered to avoid mutual interference. An ultrasonic testing machine (MSGW30, Zheda Jingyi Electromechanical Technology Co., Ltd., Hangzhou, China) is utilized to generate the T(0,1) mode wave and receive the signal from the transducers under the control of a personal computer.

### 3.2. Specimen and Loading Method

A 780 mm-long Q345 [38] steel tube with a diameter of 88.5 mm and a wall thickness of 4 mm is used as the specimen, as shown in Figure 4a. End plates with stiffeners are added to the ends of the specimen for a better force transfer. To determine the locations for transducer installations and make sure the stress between them is uniform enough for the experiment, the stress distribution is estimated by finite element simulation. The result of finite element simulation is shown in Figure 4b. It can be found that the stress keeps constant in the region 0.20 m away from the ends. As a result, the transmitting transducer is located 0.20 m away from one of the ends, and the receiving transducers are situated in the middle of the specimen and 0.20 m away from the other end, respectively. The distance between the receiving transducers is 0.19 m.

The compressive stress applied to the specimen ranges from 0 MPa to 150 MPa, which is lower than the strength of steels commonly used in structures. The loading procedure is uniformly divided into eight steps, as shown in Table 1. After every load step, when the stress keeps stable at the appointed level, ultrasonic signals are excited by the magnetostrictive transducer and received by the piezoelectric probes. This exciting–receiving process is held for 2 min to eliminate possible accidental errors. The distance, *L*, between two receiving transducers is remeasured after each load step to eliminate the distance change caused by the elastic shortening of the tube under the compressive load. Since the variation of the wave velocity caused by the acoustoelastic effect is so small, it is sensitive to the initial error of the propagation distance [33]. The axial stresses are obtained by the strain gauges. The temperature change is controlled within 0.1 °C to eliminate the influence of temperature. The above procedures are repeated for another round. The datum from the first round of experiments will be used to calibrate a stress-to-velocity relationship, and the datum from the second round of experiments will be used to verify the calibrated relationship.

### 3.3. Signal Choosing and Processing

As mentioned above, the frequency of the guided wave upon 200 kHz is not considered for mode purity. However, a relatively high frequency is beneficial for energy concentration, avoiding signal overlap and coping with the shift of the central frequency. In terms of the period of the wave, a small period makes it harder to identify the signal feature, while a larger period will cause the signal image to be too wide on the time domain and overlap with other signals. After several trial tests and comparisons, a 110 kHz, 6-period signal modulated by Hann window is selected. Its time domain diagram and frequency domain diagram are shown in Figure 5.

To get the time difference between the two received signals, their position in the time domain must be determined. Directly identifying the peak is a traditional method but may not be necessarily reliable. Thus, three other methods are considered. The first is to take the average of the zero points of the whole signal as its position, as shown in Figure 6a. The second is to calculate the centroids of each graph enclosed by the signal curve and time axis, then take the average as the signal position, as shown in Figure 6b. The third is to use the cross-correlation function to directly compare the two signals (Figure 6c). The value of the function reflects the similarity of the two signals and varies with their shift on the time space. When the value appears to be maximum, the shift is just the time difference. 

These three ways are tested on an FE model of the specimen using Comsol [39]. The solid mechanics module is used for the model. The loading of axial force and the excitation of guided waves are carried out in steady-state and transient analysis, respectively. Displacement in the axial direction is fixed on one end of the tube, and uniform axial forces are applied on the other end. Circumferential displacement is applied at the position of the transmitting transducer to excite guided waves. To ensure the accuracy of the simulation, the maximum element size is set to be less than 1/10 of the wavelength. The elastic constants of the material used in the simulations are given as *λ* = 110 GPa, *μ* = 82 GPa, *ρ* = 7850 kg/m^3^, *l* = −350 GPa, *m* = −600 GPa, and *n* = −720 GPa, based on the data of several kinds of steels measured by Takahashi [40]. In the FE simulations, the sampling rate is set to be 2 MHz, the same as that used in experiments. Because the time change in this study is smaller than the sampling period of 5 × 10^−7^ s, the obtained signal is interpolated with cubic spline when using the cross-correlation method.

At a distance of 2.5 mm, the TOFs determined by the three methods and comparisons with the theoretical result are given in Table 2. It shows that the errors of the three methods are close to each other, and all are lower than 2.0%. The zero-point method, which has the minimum error, is adopted. 

In this study, the received signal is first filtered and denoised at the central frequency of 110 kHz by wavelet transform [21] to highlight the required signal, and then its position is determined by the zero-point method.

## 4. Results

The stress to group velocity datum obtained in the first round of experiments is shown in Figure 7. According to their theoretical linear relationship given in Equation (2), a linear fitting on the datum by least squares method is carried out and yields
(5)$$V_{g}=0.0178\sigma +3474.49$$

The R-square coefficient of the linear fitting is 0.9679, which indicates an obvious linear relationship.

Equation (5) is used to predict the stresses of the same specimen based on the group velocities determined by the second round of experiments. The predicted stresses are compared with the stresses read from the strain gauges, as shown in Figure 8. It can be observed that the average error of the eight predictions is 8.2%, and the maximum error of them is 15.7%. The level of errors is comparable to previous studies on steel rods and strands [20,23,27], considering their load levels, waveguide sizes, and application scenarios. This error may root from the errors in time and length measurements. As mentioned above, there is an error of about 2% for the TOF, and this error will become more prominent because of the distortion of received signals in experiments. The propagating distance, *L*, is manually measured using a caliper with a minimum scale of 0.01 mm. Since the entire length change of the experiment is about 0.26 mm, any possible error of the length change in the range of 0.001~0.01 mm will cause some error to the predictions. Other factors, such as the electromagnetic oscillation of instruments, may also contribute to the error.

## 5. Conclusions

In this study, an axial stress measurement method for steel tubes is developed based on ultrasonic guided waves. The propagation modes of ultrasonic guided waves in a steel tube are investigated by semi-analytical finite element method. Three methods for determining the signal position in the time domain are compared in FE simulations. An experiment system is built up, and calibration and verification experiments are conducted on a tube specimen. Here are the main conclusions of the study:For tube members, the T(0,1) mode is another good choice for wave velocity measurements. Based on the numerically obtained dispersion curves, the T(0,1) mode is appropriate to be excited due to no dispersion, low propagation velocity, and consistency for different sectional sizes.The three signal positioning methods, i.e., zero-point method, graph centroid method, and cross-correlation method, showed a similar error level around 1.8%, compared to the theoretical result. They are all proved in FE simulations to be usable to determine the signal position in the time domain.There is a linear relationship between the axial stress in steel tubes and the group velocity of guided waves in steel tubes. The detailed equation of the linear relationship can be calibrated by experiments. A linear fitting with an R-square coefficient of 0.9679 is observed, and thus the linear relationship between the stress and group velocity is validated.The average error of the stresses predicted by the fitted linear equation is 8.2%, and the maximum error of them is 15.7%. The level of errors is comparable to previous studies on steel rods and strands.

It should be pointed out that the work of this study is generally a proof of concept on the guided wave-based method for axial stress measurement of steel tubes. The temperature change influences the stability of the proposed methods. However, this factor is excluded by keeping the temperature constant in this study. As mentioned above, the process of measuring the propagating distance *L* brings errors. Connecting the two transducers with a fixed-length rod will reduce the error and improve the stability of the method. Hence, more intensive studies need to be conducted to clarify the effects of some issues and then improve the accuracy and stability of the method.

## Figures and Tables

**Figure 1 sensors-22-03111-f001:**
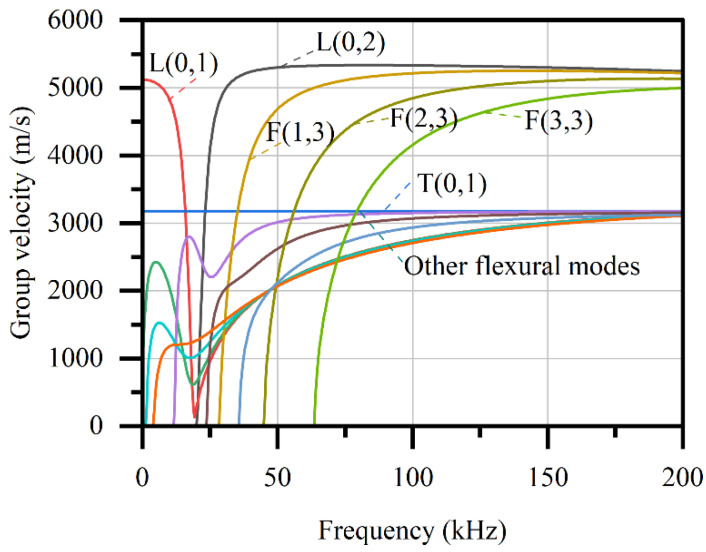
Dispersion curves of a typical steel tube with diameter of 88.5 mm and thickness of 4 mm.

**Figure 2 sensors-22-03111-f002:**
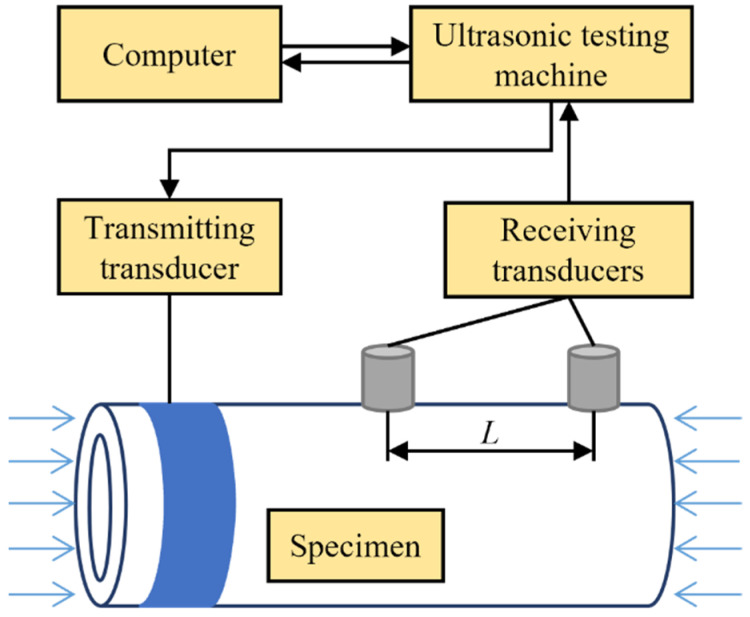
Scheme of experimental system.

**Figure 3 sensors-22-03111-f003:**
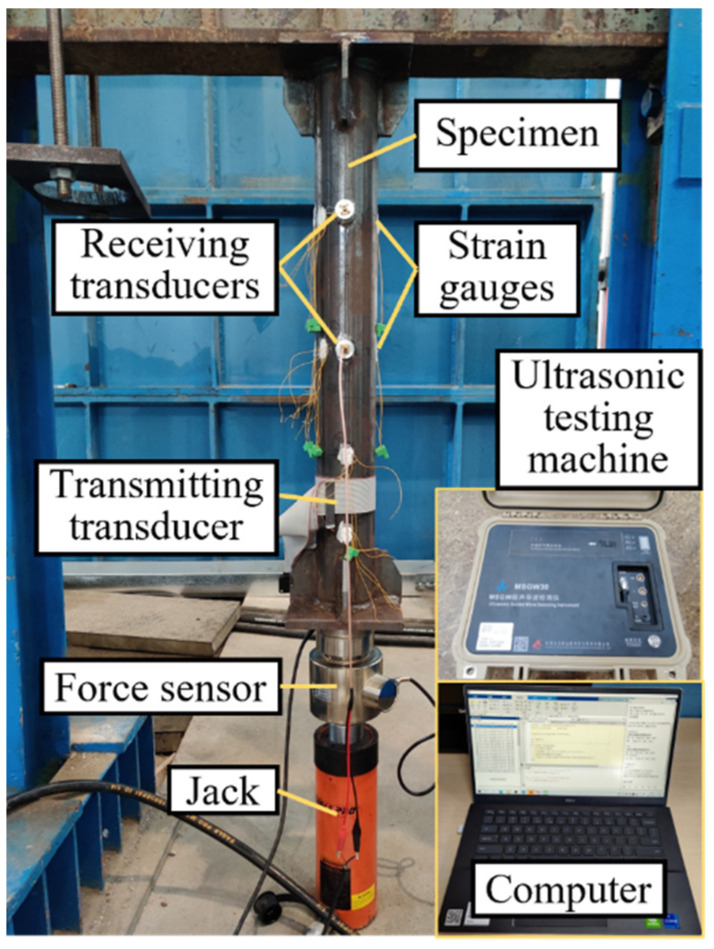
Photograph of equipment arrangements.

**Figure 4 sensors-22-03111-f004:**
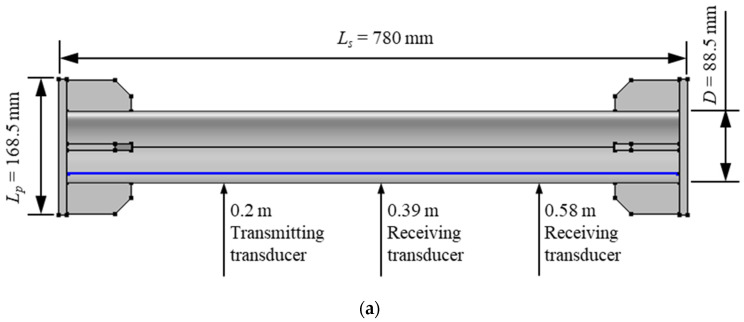
(**a**) Sizes of the specimen and (**b**) stress distribution in FE simulation.

**Figure 5 sensors-22-03111-f005:**
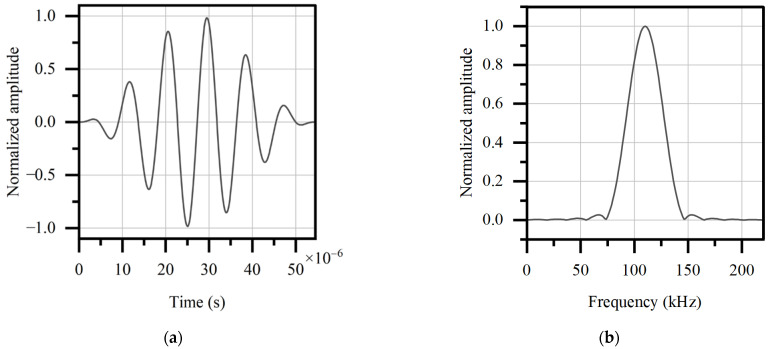
(**a**) Time domain diagram and (**b**) frequency domain diagram of excited signal.

**Figure 6 sensors-22-03111-f006:**
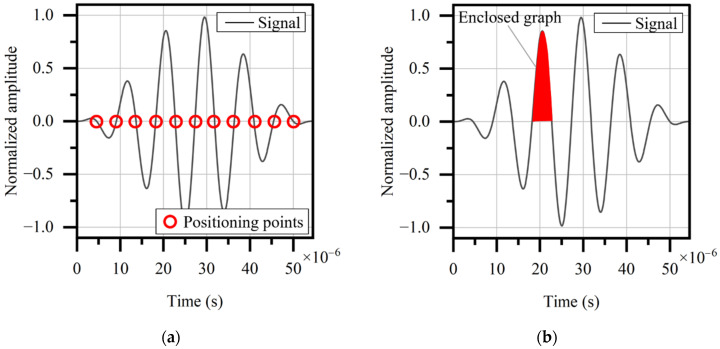
Conceptual illustration of three positioning methods: (**a**) zero-point method; (**b**) graph centroid method; (**c**) cross-correlation method.

**Figure 7 sensors-22-03111-f007:**
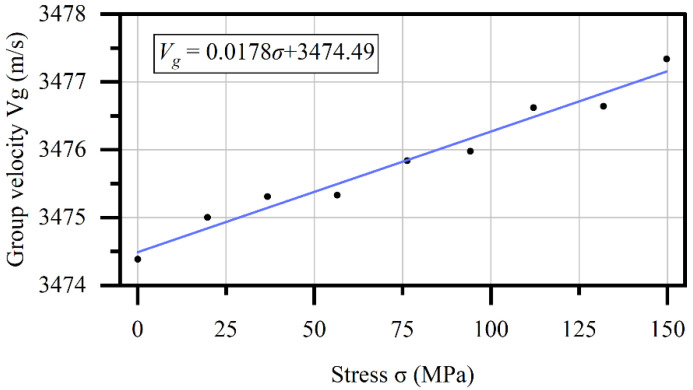
Calibration result of relationship between group velocity and stress.

**Figure 8 sensors-22-03111-f008:**
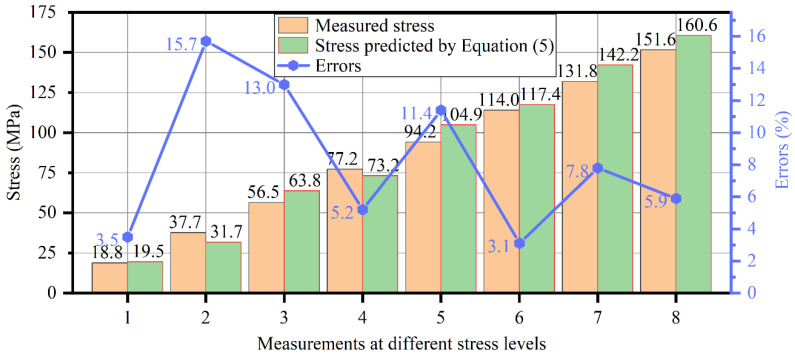
Comparison between stresses predicted by Equation (5) and those measured by strain gauges.

**Table 1 sensors-22-03111-t001:** Stress in each step of the loading procedure.

Steps	1	2	3	4	5	6	7	8
Stress (MPa)	18.75	37.5	56.25	75	93.75	112.5	131.25	150

**Table 2 sensors-22-03111-t002:** TOFs determined by different methods.

Methods	Theoretical Result	Zero-Point	Graph Centroid	Cross-Correlation
TOFs (×10^−7^ s)	7.701	7.839	7.842	7.840
Errors	-	1.79%	1.83%	1.80%

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
