# Peer review of "Axial Stress Measurement of Steel Tubes Using Ultrasonic Guided Waves"

_sensors, 2022, doi:10.3390/s22093111_

Round 1
Reviewer 1 Report
The article “Axial stress measurement of steel tubes using ultrasonic guided waves” proposes a stress measurement method for axially loaded steel tubes, based on the linear relationship between the group velocity of guided waves in the steel tube and the stress of the steel tube.
The manuscript is well designed and organized, but the scientific work and the presentation are not very clear and need improvement. Some points in the text are not well written/presented and need revision. A major revision is needed focused in the three following suggestions:
- The experimental setup is well described (please improve Fig. 3) but the FE model and simulation section provides only some basic information. The description of the model is very poor and very general and must be updated.
- The English language and style used must be improved, and grammar and spell check are required.
- The captions in all Figures are misleading and must be improved. Legends should be also added. The same holds for the Table captions.
Reviewer 2 Report
I have checked the present work. Although its experimental structure, it holds a number of motivating features. However, there are several comparable results discussed in literature and I think it deserve to be compared more properly with the outcomes of the present work. There is also one point, I think the algorithm used to address the numerical graphs must be mentioned. Moreover, the stability problem must be mentioned.
Reviewer 3 Report
This text proposed a kind of the guided-wave based method for axial stress measurement. It is interesting and maybe practical. Authors should response the following comments before the acceptance of publication.
- The author's writing should be more in line with the habit of English scientific papers. I suggest that the author should polish the full text.
- In introduction, authors should give out this method of potential applition region.
- In introduction, authors should compare the advantages and disadvantages of different methods.
- Where are the citations for Equations 3 and 4?
- All variables used by the author in the study are real numbers, and there is no imaginary number, which means that there is no dissipation of energy. In some practical cases, it is inconsistent, so the author should make this clear in the text.
- I strongly suggest that the author should carefully modify each fig, because the font and size of numbers and letters in many figs are inconsistent
Reviewer 4 Report
The authors have reported stress measurment for axially loaded steel tubes by acoustoelastic guided wave technique. My comments for the manuscript are listed below:
- The novelty is not clear. Please highlight the novelty of this manuscript in the abstract and introduction.
- The stress measurement method by acoustoelastic guided wave for axially loaded structures has been investigated. Please explain the differences with previous works in introduction.
- The introduction must be significantly improved. The literature review ignores some works regarding the guided wave theory and guided wave based stress measurement. Please also more clearly mention what benefits your proposed method has that will make it attractive for practical application. And the last paragraph of introduction must be reorganized.
- In the section 2.1, the acoustoelastic theory of ultrasonic waves propagating in a homogeneous and isotropic infinite solid is presented. However, this work is based on the acoustoelastic guided wave for stress measurement. It’s different. Please provide the relevant acoustoelastic guided wave theory.
- Due to author put forward the semi-analytical finite element method in abstract, it’s better to provide a more detailed explanation and justification for the SAFE method at section 2.2.
- The theory of acoustoelastic theory under axial stress and guided waves in tube cannot support the guided wave propagation in axially loaded tube.
- In line 136-147, this work explained why T(0,1) was suitable for stress measurement. However, there has few conncections between the proposed thoery in this work and the advantages of T(0,1). Please provide the more detailed discussion acoustoelastic guided wave propagation results.
- Please provide the detail of FE model using Comsol in line 225. How to add axial load in tube? And which the analysis models are you chose? And the influence of mesh size?
- In the experiment process, the authors selected one frequency of 200 kHz for stress measurement. It’s not sufficient to demonstrate the feasibility of the proposed approach.
Round 2
Reviewer 1 Report
The Authors edited appropriately the manuscrit following all my suggestions and in my opinion the paper can now be published.
Author Response
Thank you for your recognition of our work.
Reviewer 4 Report
I have the following comments that I would like to be addressed by the authors.
1. The literature review regarding acoustoelastic guided wave for axially loaded structures in the introduction should be improved by including some more relevant and recent citations, please try to include all the necessary references. For example:
DOI: 10.1121/1.4740491
DOI: 10.1016/j.ultras.2017.12.003
DOI: 10.1016/j.ultras.2020.106141
Author Response
Response:
Thank you for your suggestions. All the necessary references have been added to the literature review, as listed below.
Line 71-73
“Gandhi et al.[25] developed the theory of acoustoelastic Lamb wave propagation for isotropic plates subjected to a biaxial, homogeneous stress field.”
Line 78-79
“Dubuc et al.[28] explored the effect of axial stress on higher order longitudinal guided modes propagating in individual wires of seven-wire strands.”
Line 81-83
“Yang et al.[30] took temperature into account and proposed a thermo-acoustoelastic theory combined with the semi-analytical finite element method to investigate thermal effects on acoustoelastic guided wave propagation.”